# Development of Two Innovative Performance-Based Objective Measures in Feline Osteoarthritis: Their Reliability and Responsiveness to Firocoxib Analgesic Treatment

**DOI:** 10.3390/ijms231911780

**Published:** 2022-10-04

**Authors:** Aliénor Delsart, Maxim Moreau, Colombe Otis, Marilyn Frezier, Marlene Drag, Jean-Pierre Pelletier, Johanne Martel-Pelletier, Bertrand Lussier, Jérôme del Castillo, Eric Troncy

**Affiliations:** 1Groupe de Recherche en Pharmacologie Animale du Québec (GREPAQ), Department of Biomedical Sciences, Faculty of Veterinary Medicine, Université de Montréal, St.-Hyacinthe, QC J2S 2M2, Canada; 2Osteoarthritis Research Unit, University of Montreal Hospital Research Center (CRCHUM), Montréal, QC H2X 0A9, Canada; 3Boehringer Ingelheim Animal Health, Fulton, MO 65251, USA

**Keywords:** feline, osteoarthritis, firocoxib, gait analysis, stairs, performance

## Abstract

The metrological properties of two performance-based outcome measures of feline osteoarthritis (OA), namely Effort Path (Path) and Stairs Assay Compliance (Stairs), were tested. Cats naturally affected by OA (*n* = 32) were randomly distributed into four groups (A: 0.40, B: 0.25, C: 0.15, or D: 0.00 mg firocoxib/kg bodyweight) and assessed during baseline, treatment, and recovery periods. For Path, from an elevated walking platform, the cats landed on a pressure-sensitive mattress and jumped up onto a second elevated platform. Analysis included velocity, time to completion, peak vertical force (PVF), and vertical impulse. For Stairs, the number of steps and time to completion were recorded for 16 steps up and down in a 4 min period. Reliability was moderate to very good for Path and poor to good for Stairs. Different normalization methods are described in the manuscript. The placebo group remained stable within-time in Path, whereas treated cats trotted faster on the ramp (*p* < 0.0001), improved their PVF (*p* < 0.018) and completed the task quicker (*p* = 0.003). The percentage of cats completing the Stairs finish line was higher under treatment (*p* < 0.036), with huge effect size, the placebo group results being stable within-time. Both are promising performance-based outcome measures to better diagnose and manage feline OA pain.

## 1. Introduction

An outcome measure is the result of a test critical to understanding an individual’s status and progress over time. Hence, according to evidence-based medicine, outcome measures are imperative: they provide credible justification for therapeutic purposes only if they are reliable (unchanged measurement upon test and retest or across different assessors), able to demonstrate similar values across a range of individuals and eventually sensitive/responsive enough to an intervention, if any.

Among existing outcomes measurement, performance-based measures are known to evaluate how an individual performs on specific tasks, including how the task was approached [1]. As in humans [2], such clinical endpoints are of an utmost importance for veterinary medicine, particularly with the challenge that represents the detection of joint pain or disability in companion animals affected by osteoarthritis (OA). In OA cats, performance-based outcome measures are obtained through disease-specific questionnaires, or the so-called clinical metrology instruments, in which functional impairments are assessed by the owners (or veterinarians) [3,4,5]. Complementary to those subjective questionnaires, mobility or activity can be monitored in an objective manner by accelerometer-based devices [6,7]. In addition, valuable contributions to veterinary literature have been made based upon the objective analysis of ground reaction forces [4,8,9] (also called podobarometric gait analysis) in cats moving across a pressure-sensitive mattress. Performance-based outcome measures provide an overview of the cat’s overall functioning. Central sensitization may develop during OA disease progression. In addition to performance-based outcome measure, quantitative sensory tests are static or dynamic research tools that could be used to characterize the somatosensory profile of OA cats [10]. As there is no gold standard to evaluate the domains of joint pain and disability in cats with OA, there is, therefore, a fertile ground for the development of novel, reliable, and sensitive to treatment outcome measures.

Firocoxib, a COX-2 specific non-steroidal anti-inflammatory drug is commonly used to manage canine and equine OA pain (Previcox^®^ or Equioxx^®^; Boehringer Ingelheim Animal Health). Its treatment effect was mostly measured using physical examination and pain scale but also objective podobarometric (force plate) gait analysis [11,12]. Some studies investigated the use of coxibs to manage OA feline pain, but no veterinary non-steroidal anti-inflammatory drugs are currently approved in North America for the safe, long-term control of OA pain in cats [13,14]. One explanation for this delay could be the lack of objective outcome measures in the feline OA pain domain. With this thought in mind, the purpose of this study was to present the Effort Path (Path) and Stairs Assay Compliance (Stairs), two novel assessments of functional impairment in geriatric cats with OA. The aim of this study was to determine to what extent the proposed performance-based outcome measures provide reliable, homogeneous results among OA cats and their sensitivity to discriminate a dose-dependent response to an analgesic COX-2 specific treatment.

The following hypotheses will be challenged: (1) the moderate dispersion of data within successive trials (for Path) and good reliability (as mirrored by close agreement between baseline acquisition sessions) will be observed for a given animal subjected to Path and Stairs. Despite such reliability, it is assumed that the recorded outcomes will still be heterogeneous among the whole population sample. Therefore, we raised, as a second hypothesis: (2) that the uniformity of the results among cats will be improved after having normalized the data according to morphometric measurements (i.e., height, length), body weight, as well as walking velocity and time to task completion for Path. The third hypothesis concerned the sensitivity to OA therapeutics; (3) using the normalized data, Path and Stairs will be sensitive enough to discriminate a treatment effect.

## 2. Results

### 2.1. Effort Path

At the first Baseline acquisition session, cats had a mean bodyweight of 4.8 kg (3.4–6.8, Min–Max). Before jumping down, cats travelled the walking ramp at a velocity of 1.4 m/s (0.2–2.7). The coefficient of dispersion for this first outcome measure was 13.9% between trials, while it was more than twice as high (33.2%) between individuals. Outcomes recorded using the pressure-sensitive mattress are summarized in Table 1. When the cats hit the mattress, the peak vertical force (PVF) produced by each thoracic limb was higher than the mean bodyweight of the cats, reaching values higher by 123%. The inter-trials coefficients of dispersion were less than 10% for PVF and vertical impulse (VI), while they were grossly twice as high between individuals. As shown in Table 1, the values of the right and left thoracic limbs were summated for the PVF and VI outcomes, which resulted in a decrease in the dispersion of the data.

After having hit the pressure-sensitive mattress, cats get prepared to jump on the second ramp. The time necessary to perform this task, the time to completion, was considered as frames. On average, cats required 50 (25–236) frames to get off the mattress and to jump on to the second elevated ramp. The inter-trial and inter-animal coefficients of dispersion for this outcome were 12.4% and 53.5%, respectively.

The pelvic PVF produced to leave the mattress had higher coefficient of dispersion values compared to the jumping down but remained less than 20%. However, VI was the outcome with the highest coefficient of dispersion, being more than 30%. The PVF required to lift the cat was higher than its body weight, reaching values near 158%.

The vertical forces recorded for the second Baseline acquisition session as well as their spread were comparable to those obtained for the first session. The concordance of the PVF data between the first and the second Baseline acquisition sessions is presented in Figure 1. For the thoracic PVF (sum), the Spearman’s ρ was 0.80 with a coefficient of determination (R^2^) of 0.72. The dispersion of the data was slightly higher for pelvic PVF (sum) compared to thoracic PVF values corresponding to a Spearman’s ρ of 0.79 and a R^2^ of 0.67. Using intraclass coefficient correlation (ICC) index, the reliability between the acquisition sessions were as follows: values of 0.82 (95% confidence interval (CI_95_) 0.66–0.91) and 0.55 (CI_95_ 0.25–0.75) for the thoracic and pelvic PVF values, respectively. For thoracic and pelvic VI (sum), the ICC index was 0.67 (CI_95_ 0.36–0.83) and 0.63 (CI_95_ 0.35–0.81), respectively.

As an attempt to limit the dispersion of the data recorded on the pressure-sensitive mattress, selected variables (i.e., height, length, and body weight of the cat as well as velocity and time to completion) were used to normalize the PVF and VI (sum) values.

Table 2 presents raw as well as normalized, thoracic, or pelvic PVF and VI data, after having combined values of both Baseline acquisition sessions. For the PVF generated when the thoracic limbs hit the mattress, normalization by the height (ground to shoulders) of the cats reached the lowest data dispersion. For the pelvic PVF and VI, normalization by the body weight led to the lowest data dispersion.

Normalized data were used to determine a treatment effect. The placebo group was stable over time according to the univariate analysis (*p* > 0.114). A significant within-time change for the pooled treatment group was reported for the velocity on the ramp (*p* < 0.001), the time to task completion (*p* = 0.003), thoracic PVF (*p* = 0.004), pelvic PVF (*p* = 0.018), and thoracic VI (*p* = 0.002).

Cats in the pooled treatment group trotted faster on the ramp and spent less time passing the mattress at timepoints Day 15 (treatment period) and Day 24 (first recovery timepoint; Recov-1) compared to their Baseline (*p* < 0.038). Their velocity was higher on the platform at the first (Day 24) compared to Recov-2 (Day 36) timepoint (*p* = 0.016) whereas they tended to spend less time on the mattress compared to placebo group (*p* = 0.058) at Recov-1. By comparing the different dose-groups, group B had a higher platform velocity at Recov-1 compared to Baseline (*p* < 0.001) and even to Treatment (*p <* 0.001) periods. Group B showed a significant (*p* = 0.004) decrease in velocity between both recovery timepoints (see Figure 2 for the within-time velocity on the platform). On the mattress, group B cats completed the task faster at Treatment (*p* < 0.001) and Recov-1 (*p* = 0.003) compared to Baseline.

The thoracic PVF decreased in the pooled treatment group, from Baseline to Recov-1 (*p* = 0.004), and the decrease was close to statistical significance (*p* = 0.051) from Treatment to the same Recov-1. The pooled treatment group thoracic PVF was higher than for the placebo group during Treatment (*p* = 0.043) and Recov-2 (*p* = 0.034). Comparing the different dose-groups, group A cats, receiving the highest dose of firocoxib, had a decreased PVF at Recov-1 (*p* = 0.026) compared to Baseline.

The pooled treatment group of pelvic PVF increased from the Baseline to Recov-1 timepoint (*p* = 0.025) and tended to be higher at Treatment (*p* = 0.066) and Recov-2 (*p* = 0.076).

The thoracic VI, in the pooled treatment group, decreased from Baseline to Treatment (*p* = 0.021) and to Recov-1 (*p* = 0.001). Compared to Baseline, a significant decrease was reported at Recov-1 (*p* = 0.004) for group B. No difference was observed within-time or between groups for pelvic VI. The descriptive data of PVF and VI are reported in Table 3.

### 2.2. Stairs Assay Compliance

Three cats were excluded from the analysis (one was not able to complete the evaluations and two presented too much variability) during Baseline. For the remaining cats, the stairs assay compliance (Stairs) values of the first and second Baseline acquisition sessions are summarized in Table 4. In general, the twenty-nine cats climbed down fewer steps than they climbed up. The time taken to go up and down the 16-steps was similar, but the dispersion of the data was higher for the time required to go upstairs. On both Baseline acquisition sessions, the inter-individual coefficient of dispersion was elevated (18.1–25.2%) for the number of steps and (35.5–75.4%) for the time necessary to climb up or down the stairs.

ICC values for the between sessions were as follows: 0.74 (CI_95_ 0.49–0.87) and 0.68 (CI_95_ 0.31–0.86) for the number of steps up and down, respectively. For the time required to climb up and down, ICC values were 0.24 (CI_95_ −0.15–0.56) and 0.41 (CI_95_ 0.06–0.67), respectively. Normalization according to morphometric data (i.e., height, length, and body weight of the cat) were used as an attempt to limit the dispersion of inter-individual values (Table 5). A decrease of the inter-individual dispersion was found for the number of steps up, the time required to climbed up and down, with a normalization by, respectively, the length (chest to croup), the bodyweight, and the ground to elbow height measurements. No normalization improved the uniformity of the data for the number of steps down.

The finish line analysis was used as a new potent strategy to limit the data dispersion. This analysis allowed the inclusion of the two cats with the highest variability. Taking all cats performing Stairs (*n* = 31) in consideration, 32% were able to complete the finish line up and 23% the finish line down. The data dispersion associated with this distribution is summarized in Table 6. The coefficient of dispersion was improved by 3- and 17-fold, compared to the previous normalization, for the number of steps and time, respectively.

By distinguishing cats into pooled treatment and placebo groups, the percentage of cats crossing the finish line was similar between groups (*p* = 0.610) and for both Baselines (*p* = 0.097). Both Baseline-values were averaged for the following analyses. The percentage of cats reaching the finish line within-time is presented in Figure 3. The placebo group was stable within-time (*p* > 0.567). The percentage of cats crossing the finish line increased hugely in the pooled treatment group, compared to Baseline, during both Treatment (up: *p* = 0.008; down: *p* = 0.036) and Recov-2 periods (up: *p* = 0.018; down: *p* = 0.017). The number of cats completing the finish line was higher than those in the placebo group during the Treatment period (*p* = 0.032). The statistical model used was not sensitive enough to detect a dose-dependent effect (Type II error).

## 3. Discussion

About 80% of the feline geriatric population is affected by OA [15]. Despite the wide occurrence of this disease, it remains largely underdiagnosed and consequently undermanaged [16,17,18]. At the present time, there is no gold standard to objectively evaluate an OA cat’s functional (dis)abilities. Despite some feline OA questionnaires taking into consideration mobility-impaired activities, such as climbing stairs [19,20], the subjectivity in owner assessment always leads to a significant caregiver placebo effect [21].

As a comparison, humans affected by knee or hip OA can be assessed for their physical capacity according to specific objective exercises, such as the 6-min walk test, the stairs climbing test, the time up and go, and gait analysis [22,23,24]. Therefore, there is a need of performance-based objective outcome measures to better diagnose OA in cats.

Podobarometric gait analysis using a pressure-sensitive mattress has been evaluated in healthy dogs and cats [9,25] and for cats afflicted by OA [18,26]. Submitting cats to exercise before recording PVF and VI was previously reported to improve sensitivity and decrease inter-individual data variability of OA detection [27]. We kept this in mind while designing the innovative Path. Podobarometric analysis has been studied in the situation of jump reception but only in healthy cats [4]. Regarding the evaluation of cats with OA, to the best of authors’ knowledge, this is the first report of forelimbs reception and hindlimbs propulsion when jumping down and up, respectively, in movement.

Focusing on Path, to test our first hypothesis, the data dispersion and reliability between both Baseline acquisition sessions were assessed with the coefficient of dispersion and ICC. At the last phase of the jump down, we observed that the cats landed on the right or left thoracic limb first, followed by landing on the pelvic limbs. This delay, however small, led to a higher value of the first limb touching on the ground, and it often changed from one trial to another. In consequence, each thoracic limb PVF presents greater variability between trials or between animals. To limit the data dispersion, the sum of both limbs was used. In both baseline acquisition sessions, the data uniformity was greater for the thoracic or pelvic PVF (sum) values (see the coefficients of dispersion values in Table 1). Additionally, PVF values were more repeatable than VI values. This could be due to the VI calculation; it is composed of two factors, i.e., the force and the time the limb is in contact with the ground. Adding a second variable could expose the data to a greater dispersion. Despite a larger inter-individual dispersion, we noted a decrease of the coefficient of dispersion for the pelvic VI values at the second Baseline acquisition session. This could be explained by a better understanding of the Path and so a decrease in variability in the duration of the limb support. This point is in concordance with the performance-based measure definition. It evaluates how an individual performs on a specific task, here the jump, and how the task is approached. It is therefore not aberrant to think that with the habituation, the jump was apprehended with a slight difference between both Baseline sessions.

A concordance plot was used to represent the agreement of both Baseline acquisition sessions for PVF data spread. Values of thoracic PVF (Figure 1A) were moderately different from the reference line (shift about 5 degrees) and the agreement between both sessions was good with a Spearman’s ρ of 0.80 and a R^2^ = 0.72. Despite a larger spread of the pelvic PVF data (R^2^ = 0.67), the regression curve (Figure 1B) was closest to the reference line and the correlation was as good as for the thoracic PVF data, with a Spearman’s ρ of 0.79.

Whether for PVF or VI, ICC were good to very good, except for a moderate ICC for the pelvic PVF values. The latter had more inter-trial and inter-individual variability than thoracic limbs values. It could be linked to the characteristics of our cat colony; they all are affected by the OA of at least one pelvic limb joint (most often, hip) or in the lumbo-sacral vertebral axis. Therefore, the biomechanical alterations of the OA degenerative process could lead to more influence on the pelvic than thoracic limbs. Further, thoracic limbs are passive and undergo the jump, unlike the pelvic limbs which generate a force necessary for the propulsion phase. This could explain why values from pelvic limbs were higher than thoracic limbs. Nevertheless, a recent study [28] revealed that elbow and hip joints are major contributors to energy absorption during landing in healthy cats. How the impact forces are dissipated during landing for OA cats remains to be determined.

As detailed in the literature many years ago [29] and even more recently [30,31], data normalization is relevant for dog podobarometric gait analysis. The effectiveness of the normalization process for OA cats was tested in the present study. Morphometric values (i.e., length, height), bodyweight, velocity, and time to completion were variables used in the normalization process. A slight but substantial decrease in the data dispersion of thoracic PVF was obtained by a normalizing with a ground to shoulders height. Globally, the cats with the longest shoulder height had higher PVF values. As expected [29], normalizing the pelvic PVF or VI values by bodyweight led to a decrease in the coefficient of dispersion. Normalizing simply by bodyweight does not account for the possible differences in muscular mass, bony structures, or body fat. The cat’s flexibility, joint temporal, and spatial coordination, as well as the swing arm from the thoracic limbs, are also involved in the countermovement the cat executes for vertical jumping. All these factors would need to be further explored.

Therefore, the Path presented very good results for data dispersion and reliability, while normalization improved the inter-individual data variability. For the responsiveness to treatment, it is important to first highlight that the placebo group was remarkably stable within-time for all outcomes, and this helped to detect a treatment effect for the pooled treatment group. In the context of randomized controlled clinical trials, placebo responses are improvements documented in a negative control group (e.g., a group with no active intervention). The improvements can be real for the patient, such as those associated with regression-to-the-mean or a placebo by proxy (‘better care”) effect, or merely as perceived by the caregiver, such as those associated with a caregiver placebo effect, where the intense follow-up associated with a study can led to improved caregiver ratings on subjective measures or increased exchanges with the cat. In either case, real or perceived improvements in the negative control group can have a profound impact. Indeed, as previously reported on many occasions, using a subjective scale in naturally occurring OA-associated pain presented about 54 to 74% of success in cats receiving a placebo [19]. Significant within-time changes occurred in the pooled treatment group, regarding the velocity on the platform, the time to task completion, the PVF of both thoracic and pelvic limbs and the VI of thoracic limbs.

Treated cats moved faster on the platform, and they completed the task on the mattress, at the Treatment timepoint more rapidly than at Baseline. Importantly, thoracic PVF in the pooled treatment group was higher than in the placebo group at the same timepoint (after 15 days of treatment). This represents a real treatment effect. During the same timepoint, thoracic VI decreased compared to Baseline. This suggests that the OA cats were able to put more vertical force in the landing phase, while controlling it better over a shorter period of time. Finally, pelvic PVF increased compared to Baseline (close to statistical significance), suggesting the treated cats were able to put more force into propulsion.

The pooled treatment group presented a sustained beneficial effect at Recov-1, i.e., 4 days after stopping the treatment, for the velocity on the platform and the time to task completion (both were faster, compared to Baseline). The difference was no longer present at Recov-2, i.e., 3 weeks after stopping the treatment. At Recov-1, thoracic PVF was the most sensitive evaluation to detect the treatment withdrawal, as it was lower than Baseline, and apparent primarily in group A (highest dose of firocoxib). However, pelvic PVF remained higher than Baseline at Recov-1, and the effect persisted up to Recov-2.

Some dose-dependent effects were also detectable; the group A (highest dose) was the most sensitive to the treatment withdrawal (rebound effect?) and most beneficial changes observed in the pooled treatment group were attributable to group B (intermediate dose), whereas group C (lowest dose) had limited influence. Globally, the firocoxib-induced analgesia was reflected in each Path component, namely velocity on the platform and during the task on the mattress, as well as both jumping phases. Cats were able to increase their comfortable speed before jumping down, to jump with less joint impact but with more force, and to shorten their time to completion and to jump up with more propulsion force, all which reflect a better adaptability of the musculoskeletal system. Although studies have demonstrated that firocoxib improved the PVF of OA dogs [12], to our knowledge, this is the first publication supporting its efficacy in a feline OA pain complex task completion.

Although the number of stair steps climbed is impacted by chronic OA pain, no one to date has developed an objective count of stair compliance. We developed the Stairs task with the goal to monitor distance activity (number of steps up and down) and the velocity to realize the task (time up and down). The four minutes of assessment allowed us to distinguish different cat biomechanical alterations, and this delay was established after different trials to better engage the cats. Finally, to reflect about the possible fatigue felt by the OA cat completing the task, we proposed the finish line outcome.

In conjunction with positive enrichment, the Stairs task was successfully completed by 31 cats. Time up was the less reliable outcome measure, whereas, interestingly, steps up was more reliable. Cats were constant in the number of steps they climbed up but the time they spent to ascend changed. Overall, the inter-individual variability was higher for steps and time up than for steps and time down, possibly reflecting a higher panel of difficulty in climbing up than down in cat preferentially affected by hindlimb OA. The Stairs task is complex as many factors could affect the cat’s willingness to climb the stairs up and down. Therefore, the inter-individual variability was expected to be high, and also for the same cat in different sessions. The normalization process led to a better inter-individual homogeneity. Morphometric measurements (i.e., height, length) and the bodyweight influenced the spread of the Stairs values. The length impacted the number of steps to climb up; body weight had influence on the time to go up (while it influenced the pelvic values in Path), and height influenced the time to go down (as it influenced the thoracic values in Path). However, the impact of these normalizations was limited.

The finish line was used to reduce the data dispersion by distinguishing cats as responders or not. A statistically significant and huge within-time treatment effect was detectable for the pooled treatment effect (both for up and down finish line), whereas the placebo group remained remarkably stable. The finish line up outcome revealed a real treatment effect at the Treatment period, when comparing pooled treatment and placebo groups. Moreover, the within-time improvement persisted over the recovery period. The model of analysis was not sensitive enough to detect a dose-dependent effect (type II error), but a clear trend was observable under Treatment for cats completing the finish line downwards. Cats receiving the high and intermediate doses were the cats completing more stairs passages. Regarding the effectiveness of this new “Stairs Finish Line” strategy, it will be particularly interesting to determine the median number of passages for healthy cats as a potent diagnosis tool for OA cats.

For the performance-based outcomes presented herein, the remaining inter-individual variability mostly involves the pelvic limbs. Focusing on the biomechanical aspect, the kinematic analysis of the pelvic limbs of healthy dogs during ascending and descending stairs revealed the range of motion for coxofemoral, femorotibial, and tibiotarsal joints [32,33]. Pelvic limb joints were more stressed during the climbing of stairs rather than during the descent. In contrast, the thoracic limb joints of healthy dogs seem more implicated during the descent than pelvic limb joints [34]. Interestingly, this is in concordance with the best Stairs data normalization process, meaning bodyweight influences the time taken to climb stairs and the height of thoracic limbs influences the time taken to descend. Furthermore, healthy old dogs have a decreased range of motion in all joints [35]. If we extrapolate the kinetic analysis obtained with dogs to the cats, we could say that cats solicit their pelvic limbs more to ascend stairs, and OA cats have diminished their range of motion in relation probably to their joint lesions (and age?). This could lead to an increase in variability concerning the time taken for cats to ascend stairs and the number of steps up. This is true in humans, in patients affected by OA, as they display a limited range of hip joint movement, particularly when they ascend stairs [36].

Even if we validated our hypothesis for both performance-based objective outcomes, further investigations are needed to obtain valid interpretation of the related results. The validation is based on a low degree of measurement error, either systematic (corresponding to the validity) or random (corresponding to the reliability) [37]. In other words, there are three levels for an instrument measure validation: construct precision, quantification precision, and translation precision [38]. Since OA influences the gait and the cat activity, PVF will be lowest for the most affected pelvic limb(s) and the number of steps climbed will decrease while the time taken to climb increases. Moreover, bodyweight will negatively affect pelvic activities (jumping up, Stairs time up), height will influence thoracic activities (jumping down, and Stairs time down), and length of the cat will affect the number of Stairs for steps up. This information leads to the construct precision. The quantification precision is based on the reliability, which was moderate to excellent for the Path and fair to good for Stairs. It still remains to determine the translation precision by assessing healthy cats in our innovative Path and Stairs to clearly determine the difference of activity compared to cats with OA. Our study has some limitations, described as follows: due to the sample size and the multi-joint conditions of each cat, we focused on a global assessment instead of clustering them according to their affected joints. We presented two novel functional assessments, and we are aware that analyses could be refined with a distinction in groups of activity according to the joint(s) affected by OA, the magnitude of the impairment or the body condition score. Additionally, considering the importance of central sensitization during the OA chronic pain progression, the degree of nociceptive sensitization affecting OA cats could be taken into consideration in order to favor individual mechanistic-based treatment care.

Both complex behavioral Path and Stairs tasks would need to be optimized in the future. For example, the jumping up was defined at a height of 44 cm, which was easily performed by OA cats. In contrast, a jump of 100 cm was achieved by healthy cats, as reported [4,16]. The healthy cats used the flexibility of their backs to absorb kinetic energy with higher height, but the ground reaction forces increased with the jump height [39,40]. It could be interesting to see the impact of different jump heights on OA cats.

In a world of possibilities, combining the Path and Stairs tasks could be of great interest. With a fixed number of stairs passages, the velocity, time to completion, and PVF would be recorded for each passage of the staircase. This could highlight a possible correlation between the fatigue (i.e., the number of steps climbed), the mechanical forces produced at each passage and the cat’s motivation.

These new outcomes, Path and Stairs, are promising performance-based outcome measures to discriminate the dose effect of OA therapy in cats. Furthermore, the Stairs assessment requires little equipment, presents an apparent huge amplitude in analgesic responsiveness (which is a major advantage in testing new analgesic efficacy), could be optimized in the future, and even implemented in a clinic or in the cat owner’s home.

## 4. Materials and Methods

### 4.1. Animal and Housing

The study was approved by the Institutional Animal Care and Use Committee (#A176-BIA19F and #CEUA-Rech-1832). All cat care and handling adhered to the Canadian Council on Animal Care’s guidelines. Adult neutered geriatric (5.5–12.5 y) cats (*n* = 32; *n* = 16 females and *n* = 16 males) were selected based on radiographic evidence of naturally occurring OA. Hence, a radiographic screening was performed under sedation for thoracic (carpus, elbow, shoulder) and pelvic (tarsus, stifle, hip) limbs, including lumbo-sacral vertebral axis, with all requested views to confirm the diagnosis of radiographic OA. All X-rays were reviewed and scored, independently and blindly, by a Diplomate of the American College of Veterinary Surgeons (B.L.). The radiographic score corresponds to the summation of the scores (0–5) of the twelve joints evaluated. To be selected, a cat had to present some radiographic alterations (i.e., presence of osteophytes and/or subchondral sclerosis or cysts) in at least one appendicular joint to be designated as having OA. Lesions, such as meniscal mineralization or enthesiophytes, had to be associated with osteophytes and/or subchondral alteration to be radiographically significant.

Included cats were healthy (excepted the OA diagnosis) according to a physical exam and the absence of clinical and pathological findings. They were also selected based on their behavioral compliance. The cat’s characteristics (name, sex, age, number of joints affected, radiographic and body condition scores) are presented in Appendix A. Cats were part of a colony, they were group-housed in lighting-, temperature-, and humidity-controlled rooms which contained environmental enrichment (perches, covered and uncovered beds, scratching posts, and toys). Cats were fed twice daily with commercial foods according to the manufacturer’s recommendations. Fresh water was available ad libitum. Four weeks before the beginning of the experiment, cats were free of any treatment, including natural health products or veterinary diets purported to relief or ease the clinical signs of OA.

The following morphometric measurements (centimeters) were recorded for each cat: heights (ground to shoulders, ground to elbow, and ground to half of the forearm); and length (chest to croup). Cats were also weighed (kg) before each data acquisition session.

### 4.2. Treatment and Study Design

Cats were randomly distributed into four groups according to the firocoxib dose (Group A: 0.40, B: 0.25, C: 0.15 and D: 0.00 mg/kg bodyweight SID). The pooled treatment group refers to groups A, B, and C, whereas the placebo group corresponds to group D alone.

After acclimation over five weeks, cats were evaluated for three consecutive 3-week periods, i.e., twice during the baseline period for Path and Stairs, once for Path and Stairs during the treatment period with daily oral administration, and twice for Path and once for Stairs during the recovery (Recov-1 and Recov-2) period. Cats were continuously assessed by two registered veterinary technicians under the supervision of one registered doctor in veterinary medicine.

### 4.3. Effort Path

As depicted in Figure 4, the Path consisted of a custom-made initial walking ramp (427 cm long; 50 cm wide; and 100 cm high), a jump down of 56 cm, a walk over a pressure-sensitive mattress (176 cm long; 37 cm wide (Tekscan Inc, Boston, MA, USA)) and finally, a jump up of 44 cm on a second ramp. The Path was enclosed by transparent plexiglass screens to make sure the cat moved naturally and undisturbed. When the cat contacted on the pressure-sensitive mattress, a podobarometric gait analysis was performed at a resolution of 1.4 sensels/cm^2^. The calibration of the pressure-sensitive mattress was carried out at the beginning of the study according to the manufacturer’s recommendations.

Twice a week for five weeks, cats were conditioned to move freely at a comfortable velocity (average of 1.5 meters (m)/second (s)) across the Path using positive reinforcement (treats, petting, toys) during or at completion of the path.

Three to five trials were obtained for each cat. The first three valid trials (assessed after the experiment) were selected, and then averaged to characterize the gait profile for that session. A trial was considered as valid when a cat moved with regular and sufficient speed and in a straight direction with a good recording of ground reaction forces. We estimated a total per assessment test of 10 to 12 passages before collecting these three trials, and the maximum number of passages allowed for each session was 16. The timepoints of assessment for Path were Day −20 and Day −6 for Baseline, Day 15 for Treatment, Day 24 for Recov-1, and Day 36 for Recov-2.

#### Performance-Based Outcome Measures for the Effort Path

The cat’s velocity on the platform was monitored using a set of three photoelectric cells (LACIME; École de Technologie Supérieure, Montréal, QC, Canada). The velocity was expressed in meters per second (m/s).

Peak vertical force and vertical impulse were recorded for both thoracic limbs when the cats hit the pressure-sensitive mattress and for both pelvic limbs when jumping up on the second ramp. The thoracic or pelvic PVF and VI were expressed in kg and kg*s, respectively.

The number of frames between the first hit of the thoracic limb on the pressure-sensitive mattress until the last pelvic limb leaves the mattress was recorded. It corresponds to the time passing over the pressure sensitive mattress, i.e., the time of the task completion. Each frame was equal to 0.02 s. 

### 4.4. Stairs Assay Compliance

The Stairs was performed using a staircase of 16 steps (step height of 20 cm). Once a week for five weeks, cats were conditioned (clicker) to climb up and down the stair using positive reinforcement (treats, petting, toys) at the top and at the bottom of the staircase. During a four-minute period, cats were encouraged to do the maximum number of up and down steps they were able to do. The timepoints of assessment for Stairs were Day −18 and Day −4 for Baseline, Day 17 for Treatment, and Day 38 for Recov-2.

#### Performance-Based Outcome Measures for the SAC

The number of steps reached up and down, and the time (seconds) to ascend and descend the staircase were recorded for each assay during the four-minute period. During both baseline assessments, the median value of completed up and down passages for the population sample (*n* = 31) was calculated to be 7. The number of cats (percentage) in each group crossing this “finish-line” of 7 was assessed at each subsequent timepoint.

### 4.5. Statistical Analysis

For each outcome measures, the mean (Min–Max) was presented as a measure of central tendency and the interval was representative of the dispersion of the data. In addition, the coefficient of dispersion was used to evaluate the dispersion of the data [10] using the formula (1):(1)$$\frac{\sum \left|xi-Median\right|/Sample\;size}{Median}$$

For the Path inter-trials coefficient of dispersion, *xi* was the value recorded for each trial. For the inter-individual coefficient of dispersion, *xi* was the mean of the three valid trials recorded. The coefficient of dispersion was expressed in percentage (%).

A process of normalization was undertaken as a potent strategy to reduce the dispersion of the data for both performance-based outcome measures using the following variables: body weight; cats’ height and length; and velocity and number of frames (only for the Path). The normalization resulted in the original data divided by the selected variable. The potential of the normalized outcomes to reduce the data dispersion was evaluated with the coefficient of dispersion. Both Baseline acquisition sessions were combined, resulting in *n* = 64 cats for Path and *n* = 58 cats for Stairs.

The agreement between both Baseline acquisition sessions was determined according to a concordance plot and the Spearman’s ρ coefficient of correlation. The degree of reliability between measurements obtained at each session was determined using the intraclass coefficient correlation (ICC) index. The ICC index was interpreted as follows, >0.81 very good, >0.61 good, >0.41 moderate, >0.21 fair, and <0.20 poor reliability [41].

The treatment effect assessed was analyzed using a linear mixed model with time (i.e., timepoints), treatment groups, and their interaction as fixed factors. Trials were included as repeated factors. Univariate analysis was presented followed by post-hoc analysis when a statistically significant difference was observed between fixed factors. For Stairs, the finish line was analyzed using inferential analysis, Fisher test, or χ^2^ test, depending on the sample size.

## 5. Conclusions

Our study described for the first time two innovative performance-based objective outcome measures with only few acclimations in OA cats. The Path and Stairs are reliable and sensitive measurements of biomechanical alterations in OA cats. Firocoxib induced an improvement in all phases of jumping, from the velocity to the propulsion for the second ramp. The stable within-time evolution of the placebo group favored the detection of studied firocoxib therapeutic effect. The response to treatment for Group B (intermediate dose) was most apparent during the treatment and recovery periods, while for group A (high dose) the response to treatment was detected after treatment withdrawal (on Path at Recov-1). Refinement in linking OA severity to analgesic responsiveness is necessary to systematically determine a dose-dependent effect. The Path and Stairs are promising outcomes to better diagnose feline OA pain and precisely detect analgesic efficacy.

## Figures and Tables

**Figure 1 ijms-23-11780-f001:**
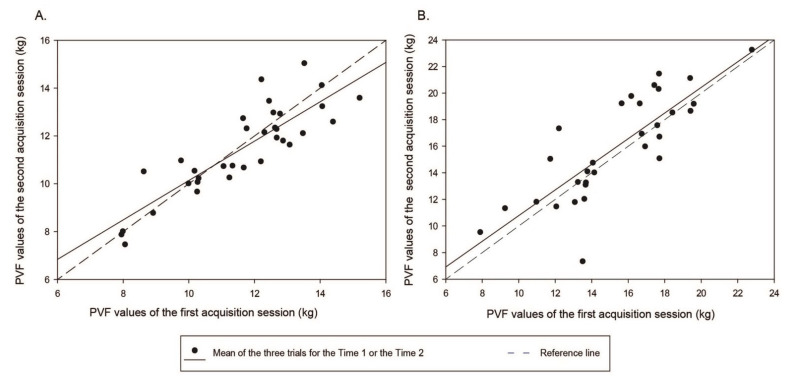
Concordance plot for the PVF values obtained during the first (*x* axis) and second (*y* axis) baseline acquisition sessions for (**A**) the thoracic limbs and (**B**) the pelvic limbs. Each point corresponds to the mean of the three valid trials for one cat. A perfect concordance is reflected by a 45° slope (dotted line).

**Figure 2 ijms-23-11780-f002:**
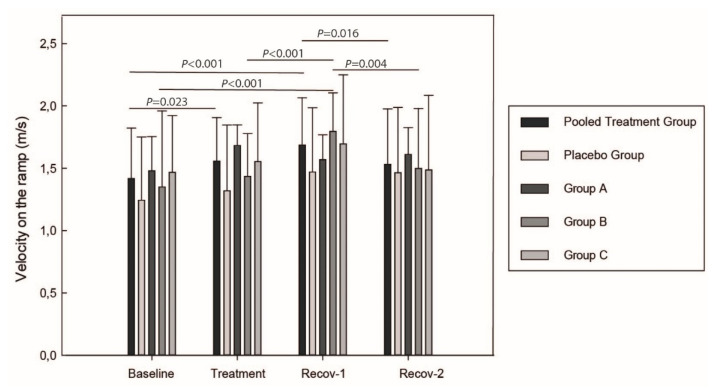
Cat velocity on the platform before jumping down. *p*-values are adjusted using Bonferroni correction. Group A = 0.40 mg/kg, Group B = 0.25 mg/kg, Group C = 0.15 mg/kg of Firocoxib. Recov-1 and Recov-2 = first and second recovery periods.

**Figure 3 ijms-23-11780-f003:**
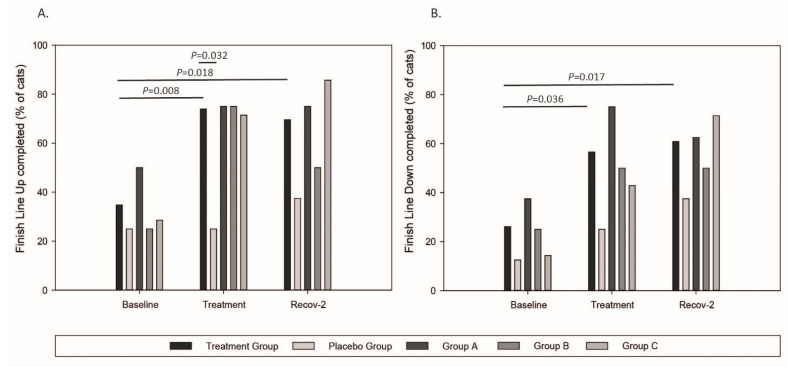
The Finish Line completed for up (**A**) or down (**B**) passages. Group A = 0.40 mg/kg, Group B = 0.25 mg/kg, Group C = 0.15 mg/kg of Firocoxib. Recov-2 = second recovery period.

**Figure 4 ijms-23-11780-f004:**
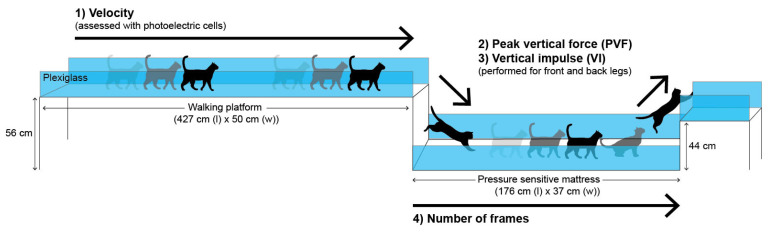
Illustration of the Effort Path. The cats walked/trotted across the walking ramp, jumped down onto a pressure-sensitive mattress and jumped up onto a raised ramp. Measurements were: (1) velocity (speed of movement across the walking platform), (2) peak vertical force (PVF), (3) vertical impulse (VI) as cats jumped down and up from the pressure sensitive mattress, and (4) the number of frames (reflecting the time to passing the pressure sensitive mattress). The entire path was enclosed by transparent plexiglass to allow cats to move naturally and undisturbed. The cats were positively motivated by rewards provided at the end of the Effort Path.

**Table 1 ijms-23-11780-t001:** Effort Path values of the first Baseline acquisition session.

	Jumping down (Thoracic Limbs)			Jumping up (Pelvic Limbs)		
Peak Vertical Force (kg)	Right	Left	Sum of Thoracic Limbs	Right	Left	Sum of Pelvic Limbs
Mean of 3 trials (Min–Max)	5.8 (3.5–8.7)	6.0 (3.1–8.7)	11.8 (6.6–15.8)	7.5 (3.7–13.4)	7.7 (3.2–12.3)	15.3 (7.7–24.0)
Coefficient of dispersion						
Inter-trials	6.9%	6.4%	**4.9%**	13.5%	10.6%	**10.2%**
Inter-individual	13.0%	14.4%	**12.3%**	19.3%	17.9%	18.5%
**Vertical impulse (kg*s)**						
Mean of 3 trials (Min–Max)	0.5 (0.3–0.8)	0.5 (0.3–1.5)	1.0 (0.7–2.2)	0.8 (0.4–6.1)	0.8 (0.2–5.1)	1.6 (0.8–11.2)
Coefficient of dispersion						
Inter-trials	8.7%	9.0%	**5.9%**	13.7%	14.6%	**12.3%**
Inter-individual	16.9%	19.5%	16.9%	37.4%	30.9%	33.1%

The peak vertical force and vertical impulse are summarized for thoracic and pelvic limbs. The coefficient of dispersion was presented to resume the dispersion of the data.

**Table 2 ijms-23-11780-t002:** Best normalization process for Effort Path values.

	Peak Vertical Force (kg)		Vertical Impulse (kg*s)	
	Thoracic Limbs	Pelvic Limbs	Thoracic Limbs	Pelvic Limbs
**Raw Data**				
Mean values (Min–Max)	11.7 (7.5–15.2)	15.5 (7.3–23.2)	1.0 (0.7–1.7)	1.5 (0.9–8.1)
Coefficient of dispersion				
Inter-individual	11.9%	21.2%	17.2%	26.0%
**Normalization Process**				
Best normalization	Ground to Shoulders	Body Weight	Body Weight	Body Weight
Coefficient of dispersion				
Inter-individual	**9.9%**	**14.6%**	**10.1%**	**21.6%**

Raw data of the thoracic or pelvic peak vertical force and vertical impulse are presented for the baseline period. The variable resulting in the best normalization, basing on the data dispersion, is presented. The coefficient of dispersion is indicated for raw data and after normalization process.

**Table 3 ijms-23-11780-t003:** Descriptive analysis of peak vertical force and vertical impulse raw data according to treatment phases.

Group	Baseline	Treatment	Recov-1	Recov-2
	Peak vertical force—Thoracic limbs (kg)			
Pooled Treatment	11.9 (8.6–15.2)	**11.7 (9.2–14.3)** ^2^	**11.2 (8.7–15.1)** ^1^	**11.4 (8.3–14.5)** ^2^
Placebo	11.1 (7.5–14.1)	10.8 (8.2–12.5)	10.9 (8.4–12.3)	10.4 (5.8–12.7)
A	11.4 (8.8–14.1)	11.1 (9.2–13.9)	10.5 (8.7–13.1) ^1^	11.1 (8.5–13.7)
B	11.7 (9.7–15.0)	12.1 (9.9–14.3)	11.6 (8.9–15.1)	11.7 (10.3–14.5)
C	12.5 [10.7–15.2]	12.1 (9.9–14.2)	11.7 (10.0–14.0)	11.6 (8.3–13.9)
	**Peak vertical force—Pelvic limbs (kg)**			
Pooled Treatment	16.1 (9.3–23.2)	16.5 (9.9–21.7)	**16.7 (9.7–23.2)** ^1^	16.4 (7.4–22.3)
Placebo	13.8 (7.3–21.1)	14.8 (6.9–21.9)	14.7 (11.2–19.2)	14.8 (8.5–21.8)
A	14.2 ]9.3–19.2)	14.6 (10.9–17.3)	14.8 (10.5–18.9)	14.9 (10.7–18.2)
B	16.8 (9.7–23.2)	17.1 (13.4–21.7)	17.6 (14.0–23.2)	16.9 (11.6–22.3)
C	17.4 (13.1–21.4)	17.8 (9.9–21.1)	17.9 (9.7–22.3)	17.3 (7.4–20.3)
	**Vertical impulse—Thoracic limbs (kg*s)**			
Pooled Treatment	1.0 (0.7–1.7)	**0.9 (0.6–1.3)** ^1^	**0.9 (0.7–1.3)** ^1^	0.9 (0.2–1.3)
Placebo	1.0 (0.8–1.4)	0.9 (0.7–1.1)	0.9 (0.7–1.2)	0.9 (0.7–1.1)
A	0.9 (0.8–1.3)	0.9 (0.6–1.2)	0.8 (0.7–1.1)	0.8 (0.7–1.2)
B	1.0 (0.7–1.5)	1.0 (0.7–1.3)	0.9 (0.7–1.3) ^1^	1.0 (0.8–1.3)
C	1.1 (0.8–1.7)	0.9 (0.7–1.2)	0.9 (0.8–1.2)	0.9 (0.2–1.1)
	**Vertical impulse—Pelvic limbs (kg*s)**			
Pooled Treatment	1.5 (0.9–5.4)	1.3 (0.9–2.0)	1.3 (0.8–1.9)	1.3 (0.3–2.1)
Placebo	1.3 (1.0–1.8)	1.3 (1.0–1.6)	1.3 (1.0–1.6)	1.3 (1.0–1.7)
A	1.4 (1.1–1.8)	1.2 (0.9–1.5)	1.2 (0.9–1.7)	1.2 (0.9–1.6)
B	1.7 (0.9–5.4)	1.4 (1.0–2.0)	1.3 (0.9–1.9)	1.4 (1.0–2.1)
C	1.5 (1.1–2.5)	1.4 (1.1–1.8)	1.4 (0.8–1.8)	1.3 (0.3–1.6)
Mean (Min–Max)				

^1^ *p* < 0.05 compared to Baseline; ^2^ *p* < 0.05 compared to placebo group. Bonferroni adjustment was used.

**Table 4 ijms-23-11780-t004:** Stairs assay compliance values for the first and second baseline acquisition sessions.

	Number of Steps		Time (s)	
	Going up	Going down	Going up	Going down
**First session**				
Mean values (Min–Max)	95 (41–144)	84 (32–112)	5 (2–22)	4 (1–9)
Coefficient of dispersion				
Inter-individual	25.2%	23.5%	75.4%	35.5%
**Second session**				
Mean values (Min–Max)	103 (32–160)	97 (32–160)	4 (2–9)	4 (2–17)
Coefficient of dispersion				
Inter-individual	18.1%	24.7%	49.7%	39.4%

The number of steps and the time to complete one passage of the Stairs at both baselines are summarized. The coefficient of dispersion reflects the dispersion of the data.

**Table 5 ijms-23-11780-t005:** Best normalization process for the stairs assay compliance values.

	Number of Steps		Time (s)	
	Going up	Going down	Going up	Going down
**Raw data**				
Mean values (Min–Max)	98 (32–160)	90 (32–160)	4.5 (1.5–22.0)	3.9 (1.3–17.0)
Coefficient of dispersion				
Inter-individual	22.6%	22.4%	60.0%	38.5%
**Normalization process**				
Best normalization	Chest to Croup	/	Bodyweight	Ground to Elbow
Coefficient of dispersion				
Inter-individual	**21.6%**	/	**55.0%**	**35.5%**

The number of steps and the time to complete one passage are summarized for the raw data and after the normalization process. The variable resulting in the best normalization, according to the coefficient of dispersion, is indicated.

**Table 6 ijms-23-11780-t006:** The Finish Line Completed descriptive analysis.

	Number of Steps		Time (s)	
	Going up	Going down	Going up	Going down
**Raw Data**				
Mean values (Min–Max)	123.6 (108.0–144.0)	120.0 (112.0–136.0)	2.7 (2.0–6.0)	2.3 (2.0–3.0)
Coefficient of dispersion				
Inter-individual	**3%**	**7%**	**22%**	**2%**

The number of steps and the time to complete one passage are summarized for cats completing the finish line. The finish line up was used to determine the going up and the finish line down allows to distinguish cats for the going down.

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
