# Peer review of "Development of Two Innovative Performance-Based Objective Measures in Feline Osteoarthritis: Their Reliability and Responsiveness to Firocoxib Analgesic Treatment"

_ijms, 2022, doi:10.3390/ijms231911780_

Round 1

Author Response

Please, see attached document.

Reviewer 2 Report

Development of two innovant performance-based objective measures in feline osteoarthritis: their reliability and responsiveness to firocoxib analgesic treatment.

The authors study the possible relationship between two methods of objective measurement of motor skills in cats with osteoarthritis tested before, after and at different stages of NSAID treatment.

The study is ambitious and interesting as, in cats in particular, the objectification of pain discomfort is particularly complicated.

The study is well conducted, with scientific rigor, however some observations are appropriate:

1) The readability of the text is not simple; it is complicated by an extensive use of acronyms and diminutives (about twenty), for some of these not correctly defined in the first use, for others defined in the captions of the images rather than in the text of the article. I advise authors to check the definition of acronyms and diminutives.

2) Cats with osteoarthritis were used but there is no reference to age, sex, origin (owner or colony?), To a remote or recent anamnesis. A summary table would improve the study.

3) Radiographs evaluated by an expert veterinarian were taken but: modality of execution of the radiographs (awake or under anesthesia? If under anesthesia with which anesthetic protocol?). How many and which joints were found positive in each cat? What is the severity of arthrosis? What is the BCS of the cats used? So: a summary table on these basic data would be very useful.

All this information is useful for the reader to better understand the results.

4) Has a clinical and neurological examination of the patients to confirm joint (or spinal radicular) pain been carried out?

5) In the introduction there are some considerations that are not related to the background but to the results or the discussion (r.75-81), perhaps they should be moved to the relevant chapter.

6) R 282: the relationship between morphometric variables and PVF and VI is still under discussion in the dog, the interdependence of some variables (weight, height, length) reduces the importance of one variable over the others, increasing the importance of the interaction between variables. I would advise the authors, in addition to what is appropriately cited (Budsberg, 1987), to consult the recent bibliography on the specific topic.

7) The effects of the study are relevant but not unequivocal, as often happens: I have not read anywhere in the article any reference to the limits of the study and the prospects for further validation and objectification tests. Don't the authors believe that some hint of limits can improve the quality of their final work?

8) (r. 386) "One m height is performed by healthy cats" this sentence seems incomprehensible, I would ask the authors to reformulate the text.

For the above considerations, I suggest that the authors review the article and make the recommended changes in order to make it eligible for publication.

Author Response

Please, see attached document

Round 2

Reviewer 2 Report

Dear Authors,

the modified version, including suggestions and requests for changes, appears complete. I have no other comments and, in my opinion, the article is adequate for publication